# Design of Functional Polymer Systems to Optimize the Filler Retention in Obtaining Cellulosic Substrates with Improved Properties

**DOI:** 10.3390/ma16051904

**Published:** 2023-02-25

**Authors:** Elena Ungureanu, Maria E. Fortună, Denis C. Țopa, Andrei Lobiuc, Ovidiu C. Ungureanu, Doina C. Jităreanu

**Affiliations:** 1Department of Plant Science, “Ion Ionescu de la Brad” University of Life Sciences Iasi, 3 Mihail Sadoveanu Alley, 700490 Iasi, Romania; 2Inorganic Polymers Department, “Petru Poni” Institute of Macromolecular Chemistry, Grigore Ghica Voda Alley 41A, 700487 Iasi, Romania; 3Department of Biomedical Sciences, “Stefan cel Mare” University, 13 Universitatii Str., 720229 Suceava, Romania; 4Department of Biology and Life Sciences, “Vasile Goldis” Western University of Arad, 94 The Boulevard of the Revolution, 310025 Arad, Romania

**Keywords:** calcium carbonate precipitated (PCC), cationic polyacrylamide (cPAM), cellulosic substrates, filler, mechanical and optical properties, polyDADMAC

## Abstract

In the present work, the possibility of increasing the calcium carbonate (CaCO_3_) content in sheets of paper to optimize their properties was investigated. A new class of polymeric additives for papermaking is proposed as well as a method for their use in paper sheet containing the CaCO_3_ precipitated addition. Calcium carbonate precipitated (PCC) and fibers cellulose were adjusted with a cationic polyacrylamide flocculating agent (polydiallyldimethylammonium chloride (plyDADMAC) or cationic polyacrylamide (cPAM)). PCC was obtained in the laboratory by a double-exchange reaction between calcium chloride (CaCl_2_) and sodium carbonate (Na_2_CO_3_) suspension. After testing, the dosage of PCC was established at 35%. To improve the systems of additives studied, the materials obtained were characterized and their optical and mechanical properties were analysed. The PCC had a positive influence over all of the paper samples, but in the case of use of cPAM and polyDADMAC polymers the paper obtained had superior properties compared to the paper obtained without additives. Also, the samples obtained in the presence of cationic polyacrylamide exhibit superior properties to those obtained in the presence of polyDADMAC.

## 1. Introduction

The use of methods of filling cellulose fiber paper with calcium carbonate precipitate registered a continuous increase [1,2]. The initial aim was reduce the cost, to improve the printing properties of high-fill paper, decreased energy consumption, improvement in machine speed [3,4]. However, at a high content of filler material in the paper, the strength can reduced due to the interference of the inter fiber bonding [5]. The filler can be added to a cellulosic substrate by different methods. By conventional procedure the filler are incorporated directly into the fiber stock suspension and the mineral is retained in the paper either by mechanical filtration or by use of retention aid. The additives can be used in paper manufacture as agents for improving retention, paper machine operation and the strength properties of the final product [6].

For enhancing paper quality, filler loading is performed which, in its conventional setting, means dosage of filler particles and their uniform incorporation into the suspension of fibers. However, in these conventional methods, fillers aren’t always well retained into the paper sheet as the filler particles start forming strands, leading to lower efficiency and high residual levels of filler into wastewater. A method [1,7] developed to overcome these problems was to directly precipitate calcium carbonate within pulp fibers walls and lumen. Using this method [1,8], paper strength was reduced compared with conventionally loading paper with filler. Furthermore, a decrease in the environmental impact was obtained when combinations of different additives were used, by reducing the contamination of the process water due to high filler retention and also reducing energy requirements for pulp stock [1,4].

Further improvements of paper quality can be obtained when calcium carbonate (CaCO_3_) is in a definite crystal form, and superior brightness and durability of the end product can be obtained using well defined particle size and distribution. Calcium carbonate precipitate (PCC) offers traits, such as reduced particle size, high purity, tight particle size distribution, regular crystal shape, compared to unmodified calcium carbonate [9,10]. While PCC exists in 3 basic forms, calcite, aragonite, and vaterite [11], the properties induced by calcium carbonate filler depend on the quality of water used in suspensions, on dissolved and colloidal substances and on the retention polymer added [10]. Additives can be used in paper manufacture to control the particle size and morphology of PCC [11], as well as agents for improving retention, paper machine operation and optical and strength properties of the product [12,13]. 

A mechanism for explaining the properties of additives is the difference of potential between the negatively charged fibers and the positively charged additive polymers which interact due to electrostatic forces [14,15]. While filler materials can improve paper properties such as opacity other properties such as strength and brightness [16] can be reduced. As such additives should be used in a way that paper properties are kept or enhanced. To accomplish such goals fillers should have a high ability to bond to fibers, a property less common in their natural forms therefore chemical modification is required. In the meantime, additives should be as environmentally friendly as possible therefore new and renewable or low-cost materials are sought after [17,18,19]. An example of such material is lignin, a polymer which is widely available in natural sources, or often as a byproduct in economical processes. When developing a new polymer as an paper additive, observing the synergism or way of interaction of its pigments or other molecules with fibers is the probable way to design high performance additives [20]. Polymers, such as polyDADMAC and cPAM, are currently introduced as retention additives in paper production. Optimized paper properties were shown to be obtained using both pigments and cellulosic materials [20,21,22]. One cationic polymer with significant flocculant properties and good performance is polyDADMAC, which is also non-toxic, while also aiding in papermaking sewage treatment reduction costs. Flocculation by polyDADMAC is due to bridging adsorption between polymer and suspended colloidal particle. One mode of action of polyDADMAC is explained by the ability to neutralize negative surface charges, thus acting as an effective additive [23,24].

Polyacrylamide is the most usually water-soluble synthetic retention polymer used in the paper industry. Polyacrylamides can be homopolymers or copolymers, nonionic, anionic or cationic polymers. In the paper manufacturing, polyacrylamide has a more of uses according to the different molecular weight (dispersant, reinforcer, retention and filtration aid, flocculants in papermaking wastewater treatment). cPAM flocculates of CaCO_3_ even in the presence of large excess of dissolved anionic such as sulfonated kraft lignin. It representing the typical retention aid, acts by bridging mechanism [25,26].

Both polyDADMAC and cPAM cationic polymers are used to improve retention of fine particles in papermaking. They can affect the PCC fillers and can affect the interaction of these with pulp fibers in different mods [27,28]. 

The objective of this work was to analyze and optimize the efficiency of fillers using calcium carbonate precipitated by double exchange reactions in combination with cationic polymers which are used to improve retention of fine particles in papermaking.

A single-component system consisting of two additives (polyDADMAC and cPAM polymers) was designed, which proved to be efficiency alternative for calcium carbonate fillers deposition on fibers. The optimum percent of polymers, their behavior and the impact over optical and mechanical paper properties were analyzed. Cationic polymers added in the paper pulp offer positive effects on retention, dehydration and cellulosic substrate properties.

## 2. Experimental

### 2.1. Materials

Kraft pulp fibers from Braila, Romania; CaCl_2_ (96%) and Na_2_CO_3_ (99.8%) from Chemical Company, Iasi were used to obtained PCC. Retention additives: cationic polyacrylamide (cPAM) Ciba^®^ Percol^®^ Co 455 (charge density = 3% and mass ~4–10 million Daltons) were acquired from CIBA Specialty Chemicals Canada Inc; poly(diallyldimethylammonium chloride) as Alcofix^®^ 169 (polyDADMAC), Sigma-Aldrich, Germany (0.5 g/L).

### 2.2. Methods

To determine refining degree was used the Schopper-Riegler instrumentation (Hangzhou, Zhejiang, China) and 30° SR was obtained for all samples studied.

Paper sheets obtained with weight of 70 g/m^2^ were conditioned (24 h at 23 °C and 50% RH) and analyzed.

CaCO_3_ was measured according to the Tappi Standard—T413 (Tappi Test Methods, 1999). The CaCO_3_ content is calculated from the ash content in the hand sheets, taking into account the incineration loss of the precipitated calcium carbonate.

Air permeability, measured by the DL-WEB apparatus at a 1 kPa pressure, Guangdong, China.

Opacity and brightness (according to ISO 2471), were determined by the spectrophotometer L&W Elrepho 2000 apparatus, Kista, Sweden.

Strength properties: breaking length (km) measured by Instron apparatus, according to ISO 1924 and burst factor (kPa·m^2^/g) measured by the Schopper-Dale apparatus, according to ISO 2758, Bucuresti, Romania.

Scanning electron microscope (SEM) images were obtained with VEGA/TESCAN instrument, Netzsch, Germany.

X-ray diffraction was determined on a D8 ADVANCE, Bruker-AXS apparatus, (Billerica, MA, USA).

Thermogravimetric analyses (TG and DTG) were obtained using STA 449F1 Jupiter equipment (Netzsch, Selb, Germany). 

### 2.3. Experimental Procedure

Precipitated calcium carbonate can be produced as a heterogeneous liquid-solid mixture, following the very rapid precipitation reaction of two aqueous solutions of carbonate salts by the next reaction:CaCl_2_ + Na_2_CO_3_→CaCO_3_ + 2NaCl(1)

For this purpose, PCC was added to pulp suspension under stirring. After more retention tests the content of calcium carbonate was at 35%. The paper pulp was treated with cationic polyacrylamide or polyDADMAC under stirring. After several tests, it was found that it is more efficient to add cationic polymers after adding the CaCl_2_ solution and before adding the Na_2_CO_3_ solution.

For the polyDADMAC samples the added doses were: 0%, 0.2%, 0.4%, 0.8% and 1% *w*/*w* on pulp and it was added to the suspension of pulp with 1% consistency and was allowed to stir during 10 min.

For the cPAM samples the added doses were: 0%, 0.02%, 0.04%, 0.08% and 0.1% *w*/*w* on pulp and also as in the case of using polyDADMAC, it was added to the suspension of pulp with 1% consistency and was allowed to stirring during 10 min.

The paper sheets were obtained and for each samples, three variants were used for replication:pulp fibers/35% PCCpulp fibers/35% PCC/polyDADMAC (0%, 0.2%, 0.4%, 0.8% and 1%)pulp fibers/35% PCC/cPAM (0%, 0.02%, 0.04%, 0.08% and 0.1%).

According to Figure 1, the experimental design includes obtaining fibrous pulps filled with calcium carbonate precipitated and preparing paper sheets from these pulps, as well as forming paper sheets by using additives.

## 3. Results and Discussion

### 3.1. Effects of Polymers on Paper Stock Properties

The results obtained shown that it is possible to obtain pulps with 11–14% of CaCO_3_ content, which thus can be 100% retained in paper sheet that is formed of these pulps. The content of PCC increase with the addition of cationic polymers.

The porosity of fibers loading with PPC and polymers was represented by air permeability and it was found that this was influenced by the type of fiber used, the content of filling material, as well as by the type and doses of retention additives used.

The results from Table 1 show that air permeability increases with increasing CaCO_3_ content. Although the calcium carbonate content registers very close increases for the both additives (polyDADMAC and cPAM), the air permeability is higher for cPAM, which shows that cPAM has a stronger influence on the paper structures.

From Figure 1a,b result that the addition of the additives, polyDADMAC and cPAM have positive effects on the retention of the filling material, as well as on the dehydrating capacity of the pulp.

The both additives behave similarly in terms of dehydration time. Calcium carbonateretention increases with the addition of retention additives. For cPAM, PCC retention increases by 6.5 units comparatively with the system using polyDADMAC where PCC retention increases by only 4.5 units.

### 3.2. X-Ray Diffraction

The presence of PCC in the fibers and the polymorphic form of PCC particles retained in paper pulp were determinates by X-ray diffraction analysis. Figure 2 presents the X-ray diffraction results for PCC and for paper pulp loaded with PCC. X-ray diffraction of the PCC and of the fibers analysis had shown a characteristic calcite diffraction model. In Figure 2, the peak at 29.3° (104) has shown the calcium carbonate content for calcite, and for cellulose it is show at 22.5° (002).

By adding polymers (polyDADMAC and cPAM), no significant differences are observed. 

### 3.3. Thermogravimetric Analysis

Thermal degradation properties (DTG and TG) of PCC samples were investigated. The weight losses as a function of temperature for all samples are shown in Figure 3. 

The PPC decomposition behavior at 850 °C, 20 min in pure N_2_ atmosphere (Figure 3a) reveals similar processes of samples decomposition. Up to 4 phases including weight loss are recorded for samples, but there only one weight loss phase occurs during decomposition of calcium carbonate at 550–830 °C. The conversion of PCC to CaO is indicated by weight loss, usually occuring rapidly in the range 550 to 830 °C. After than 850 °C, no further significant weight loss was observed, thus, this temperature was chosen as holding temperature to investigate the effect of calcination dwelling time. All materials recorded up to 44% weight loss and a residual ~55 wt% of converted CaO, thus resulting a conversion factor of ~ 0.55 gCaO/gCaCO_3_. 

Thermogravimetric characteristics of the samples are presented in Table 2.

DTG analyses (Figure 3b) showed that PPC displays only one peak corresponding to substance decarbonation, whereas, by polymer addition, the decomposition peak was shifted to a higher temperature value.

### 3.4. Scanning Electron Microscopic Images (SEM)

The morphological characteristics of PCC particles deposited onto the cellulose fibers were investigated by SEM. 

From SEM images (Figure 4a–c) we can observe the crystallization of PCC which take place with the formation of calcite microcrystals of specific shape. 

The polymers are most widely used in control the size of the particles and morphology of PCC [29]. Due superior optical properties, the scalenohedral calcite has reached like most used PCC filler in papermaking. 

The SEM images (Figure 5a–d) of the paper samples showed that they are obtained PCC calcite microcrystals of specific shape and also revealed that the largest number visible surface crystals of PCC show up for 35% PCC/1% polyDADMAC sample.

### 3.5. Optical Properties

A series of factors as particle size, particle shape and distribution of the PCC contribute to the optical and mechanical properties of the paper. The efficiency of the production of writing/printing paper depends most of the time on the solutions used to obtain the optical properties sought. This aspect is very important because some factors that have a positive effect on the degree of whiteness and opacity can negatively influence the mechanical properties such as the strength of the product.

Both polymers used led to small variations in the filling material content, for close values of the degree of whiteness and opacity. Table 3 shows the optical properties and the average deviation calculated for the both cases studied. The influence of polyDADMAC and cPAM cationic polymers on brightness and opacity properties of the paper loaded with PPC are presented in Figure 6a,b.

From Figure 6 we can see that the highest opacity is obtained in the case of loaded sample with PCC/cPAM. This can be explaining due the presence of PCC particles at the fibre surface, as well as within the fibre wall pores which are well dispersed. This is also evident from the SEM images by the smaller PCC particle size. The lower opacity of the PCC sample without the addition of polymers can be explain by non-uniformity disppersion of the PCC particles into the paper structure.

### 3.6. Physical-Mechanical Properties of Paper Samples

For a good evaluation of the effectiveness of cationic polymers on the strength properties, the breaking length of the paper was calculated taking into account its calcium carbonate content (35%). At variable addition of polyDADMAC and cPAM the breaking length of the paper filled with 35% CaCO_3_ shows higher values with increasing additive addition (Figure 7). Also the PCC content in the paper registers a greater increase in the case of the cPAM sample and the breaking length is superior in this case, compared to the sample with polyDADMAC where statistically insignificant variations are observed.

The burst index does not register important changes with the addition of additives, as can be seen from Figure 8. The evolution of the burst index for the sample with polyDADMAC is comparable to cPAM sample.

## 4. Conclusions

The retention has been found to be a critical aspect of the manufacturing process of writing/printing papers with alkaline media, so by testing different chemical additives at different additions (polyDADMAC and cPAM), improvement of the retention of the filler material was achieved: cPAM gives the best results in terms of CaCO_3_ retention, but the optical properties values are similar with other treatments; air permeability is better when cPAM is used as an additive compared to the polyDADMAC system. The strength properties of the paper filled with 35% CaCO_3_ increase with the addition precipitate of the additive, for both types of additives used.

The effects of the PCC on paper properties showed a greater stabilization when a cationic polymers was added.

## Data Availability

Data is contained within the article.

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
