# Peer review of "Design of Functional Polymer Systems to Optimize the Filler Retention in Obtaining Cellulosic Substrates with Improved Properties"

_materials, 2023, doi:10.3390/ma16051904_

Round 1

Reviewer 1 Report

Manuscript reports on a study on the effect of polymer additives on properties of calcium carbonate filled paper. This can be interesting to those working in the area of “paper materials”, consequently, for readership of Materials journal. Aim is clearly stated, methodology is standard and adequate to fulfill the goals. I place several comments to be considered before the final acceptance:

The part of manuscript text referring on Fig. 3 speaks about weight losses. However, Fig. 3a actually shows increased weight in the range about 150-450 °C. Authors should comment on this and add ore details on TG measurements in the methods section (heating rate, atmosphere etc.).

Is the crystalline structure/type of calcium carbonate important for the filler application? Or are the XRD results presented just as a standard information on the precipitation product?

Introduction informs about related works (papers). I would be nice to read somewhere at the end how the results of authors´ work compare with previous work and what exactly new they bring.

Author Response

We would like to thank the reviewer for all valuable comments and remarks which helps us to really improve the scientific quality of our revised manuscript. We have carefully considered all the reviewer comments and have revised the manuscript in light of them. The suggested modifications were clearly marked in red in the revised manuscript. Details of our responses to each reviewer comment are shown below. We hope you will find these revisions rise to your expectations.

Reviewer 2 Report

Dear Colleagues,

comments are presented in the text of the article.

Best regards

Author Response

We would like to thank the reviewer for all valuable comments and remarks which helps us to really improve the scientific quality of our revised manuscript. We have carefully considered all the reviewer comments and have revised the manuscript in light of them. The suggested modifications were clearly marked in green and red in the revised manuscript. Details of our responses to each reviewer comment are shown below. We hope you will find these revisions rise to your expectations.

Round 2

Reviewer 2 Report

Thanks to the authors for their comprehension and their scrupulous answers to all comments.

Minor editing is required without sending the result to the reviewer. The values of the content of calcium carbonate in Fig. 7 and Fig 8 should be indicated with the same accuracy, for example, to two or to three significant figures. You can generally not give these values near each point, as it is done in Fig.6. They are visible on the right Y-axis.

Best regards

Author Response

We thank again Reviewer for all important comments which helped us to really improve the scientific quality of our revised manuscript. 

Best regards,

The Authors